# FEDERATED REPRESENTATION LEARNING VIA MAXIMAL CODING RATE REDUCTION

## ABSTRACT

We propose a federated methodology to learn low-dimensional representations from a dataset that is distributed among several clients. In particular, we move away from the commonly-used cross-entropy loss in federated learning, and seek to learn shared low-dimensional representations of the data in a decentralized manner via the principle of maximal coding rate reduction ($MCR^2$). Our proposed method, which we refer to as FLOW, utilizes $MCR^2$ as the objective of choice, hence resulting in representations that are both between-class discriminative and within-class compressible. We theoretically show that our distributed algorithm achieves a first-order stationary point. Moreover, we demonstrate, via numerical experiments, the utility of the learned low-dimensional representations.

## 1 INTRODUCTION

Federated Learning (FL) has become the tool of choice when seeking to learn from distributed data. As opposed to a centralized setting where data are concentrated in a single node, FL allows datasets to be distributed among a set of clients. This subtle difference plays an important role in practice, where data collection has moved to the edge (e.g., cellphones, cameras, sensors, etc.), and centralizing all the available data might not be possible due to privacy constraints and hardware limitations. Moreover, under the FL paradigm, clients are required to train on their local datasets, which unlike the centralized setting, successfully exploits the existence of available computing resources at the edge (i.e., at each client).

The key challenges in FL include dealing with (i) data imbalances between clients, (i) unreliable connections between the server and the clients, (iii) a large number of clients participating in the communication, and (iv) objective mismatch between clients. A vast amount of successful work has been done to deal with challenges (i), (ii), and (iii). However, the often-overlooked challenge of objective mismatch plays a fundamental role in any distributed problem. For an client to participate in a collaborative training process (as opposed to training on its own private dataset), there must be a motivation: each client should see itself improved by taking part in the collaboration. Recent work has shown that even in the case of convex losses, FL converges to a stationary point from a mismatched optimization problem. This implies that there are cases where certain clients own the majority of the data (or even of certain classes), and see their individual performance curtailed by the collaborative approach.

When optimizing the average of the losses over the clients, the solution to the optimization problem generally differs from the solution of the individual per-client optimization problems. Objective mismatch becomes a particularly difficult problem in FL given the privacy limitations, which prevents the central server from curtailing this undesirable effect. Moreover, given that in standard FL, the central server possesses no data, and that no proxies of data structures should be shared, a centralized solution cannot be implemented. In order to resolve the objective mismatch issue, several approaches have been proposed. However, most such approaches rely on obtaining more trustworthy gradients in the clients, at the expense of either more communications rounds, or more expensive communications.

In this work, we propose an alternative representation learning-based approach to resolve objective mismatch, where low-dimensional representations of the data are learned in a distributed manner. We specifically bridge two seemingly disconnected fields, namely federated representation learning and rate distortion theory. We leverage the rate distortion theory to propose a principled way of

optimizing the coding rate of the data between the clients, which does not require sharing data between clients, and can be implemented in the standard FL setting, i.e., by sharing the weights of the underlying backbone (i.e., feature extractor) parameterizations. Our approach is collaborative in that all clients are individually rewarded by participating in the common optimization objective, and follows the FL paradigm, in which only gradients of the objective function with respect to the backbone parameters (or equivalently, the backbone parameters themselves) are shared between the clients and the central server.

**Related Work.** Several studies have been conducted in the context of FL to show the problem of objective mismatch, by proposing modifications in the FL algorithm (Yang et al., 2019), adding constraints to the optimization problem (Shen et al., 2021), or even including extra rounds of communication (Mitra et al., 2021). As opposed to these methods, we propose to tackle the problem by introducing a common loss that is in all clients' self-interest to minimize. Another line of research seeks to learn personalized FL solutions by partitioning the set of learnable parameters into two parts, a common part, called the backbone, and a personalized part, called the head, to be used for individual downstream tasks. Often referred to as personalized FL, this area of research is interested in learning models utilizing a common backbone that is collaboratively learned among all clients, while personalizing the head to each individual agent's task or data distribution Liang et al. (2020); Collins et al. (2021); Oh et al. (2021); Chen & Chao (2021); Silva et al. (2022); Collins et al. (2022); Chen et al. (2022). We, on the other hand, are interested in learning representations in a principled and interpretable way, as opposed to converging to a solution without any guarantees on its behavior. In the context of information theory, rate distortion theory has been used to provide theoretical (Altuğ et al., 2013; Unal & Wagner, 2017; Mahmood & Wagner, 2022) and empirical (Ma et al., 2007; Wagner & Ballé, 2021) results on the tradeoff between the compression rate of a random variable and its reconstruction error. However, most such solutions are centralized.

**Contributions.** We summarize our key contributions as follows:

1. We introduce a theoretically-grounded federated representation learning objective, referred to as the maximal coding rate reduction ($MCR^2$), that seeks to minimize the number of bits needed to compress random representations up to a bounded reconstruction error.

2. We demonstrate that obtaining low-dimensional representations using our proposed method, which we refer to as `FLOW`, entails an objective that is naturally collaborative, i.e., all clients have a motivation to participate in the learning process.

## 2 BACKGROUND

### 2.1 FEDERATED LEARNING

Consider a federated learning (FL) setup with a central server and $N$ clients. For any positive integer $M$, let $[M]$ denote the set $\{1, \ldots, M\}$ containing the positive integers up to (and including) $M$. Each client $n \in [N]$ is assumed to host a local dataset of labeled samples, denoted by $\mathcal{D}_n = \{(x_i^n, y_i^n)\}_{i=1}^{|\mathcal{D}_n|}$, where $x_i^n \in \mathbb{R}^D$ and $y_i^n \in [K]$, $\forall i \in [|\mathcal{D}_n|], \forall n \in [N]$. Focusing on a set of parameters $\theta \in \Theta$, we assume that the $n^{\text{th}}$ client intends to minimize a local objective, denoted by $f_n(\mathcal{D}_n; \theta)$, given its local dataset $\mathcal{D}_n$. In many cases, such as the cross-entropy loss (CE), this local objective can be decomposed as an empirical average of the per-sample losses, i.e.,

$$f_n(\mathcal{D}_n; \theta) = \frac{1}{|\mathcal{D}_n|} \sum_{n=1}^{\mathcal{D}_n} \ell(h_\theta(x_i^n), y_i^n), \tag{1}$$

where $h_\theta : \mathbb{R}^D \to [K]$ is a parameterized model that maps each input sample $x$ to its predicted label $h_\theta(x)$, and $l : [K] \times [K] \to \mathbb{R}$ denotes a per-sample loss function.

The global objective in the FL setup is to find a single set of parameters $\theta^*$ that minimizes the average of the per-client objectives, i.e.,

$$\theta^* = \arg\min_{\theta \in \Theta} \frac{1}{N} \sum_{n=1}^{N} f_n(\mathcal{D}_n; \theta). \tag{2}$$

It is assumed that the clients in a FL setup cannot share their local datasets with each other. This implies that the optimization problem in (2) needs to be solved in a distributed manner. To that end, we assume that each client $n \in [N]$ maintains a *local* set of parameters $\theta_t^n \in \Theta$ over a series of time steps $t \in [T]$. Each client performs $\tau$ number of local updates using stochastic gradient descent (SGD), and then the local parameters are sent to a central server every $\tau$ time steps, so that the server averages clients' parameters and broadcasts the resulting aggregated parameters to to the clients to replace their local models. More precisely, denoting the learning rate by $\eta$, and letting $\hat{\nabla}_\theta$ represent the stochastic gradient with respect to the model parameters, the sequential parameter updates are given by

$$\theta_{t+1}^n = \begin{cases} \theta_t^n - \eta \hat{\nabla}_\theta f_n(\mathcal{D}_n; \theta_t^n) & \text{if } t \bmod \tau \neq 0, \\ \frac{1}{N} \sum_{n=1}^N \theta_t^n & \text{o.w.} \end{cases} \tag{3}$$

This forms the basis of the FedAvg algorithm (McMahan et al., 2017).

### 2.1.1 PERSONALIZED FEDERATED LEARNING

Leveraging the representation learning paradigm (Bengio et al., 2013; Oord et al., 2018; Chen et al., 2020), the parameterized model $h_\theta : \mathbb{R}_D \to [K]$ can be decomposed into two components, namely i) a *backbone* $h_\phi : \mathbb{R}^D \to \mathbb{R}^d$, parameterized by a set of parameters $\phi \in \Phi$, that maps each input sample $x \in \mathbb{R}^D$ to a low-dimensional *representation* $z = h_\phi(x) \in \mathbb{R}^d$, where we assume that $d \ll D$, and ii) a *head* $h_\psi : \mathbb{R}^d \to [K]$, parameterized by a set of parameters $\psi \in \Psi$, that maps the representation $z \in \mathbb{R}^d$ to the predicted class $h_\psi(z) = h_\psi(h_\phi(x)) = h_\theta(x) \in [K]$. This implies that the set of end-to-end model parameters is given by $\theta = (\phi, \psi)$, with the corresponding parameter space being decomposed as $\Theta = \Phi \times \Psi$.

Such a decomposition can then be used to train a *shared backbone* for all the clients using the FL procedure, while the training process for the head can be *personalized* and local for each client. In particular, for the $n^{\text{th}}$ client, assume that the local objective $f_n(\mathcal{D}_n; \theta)$ can be decomposed into an objective on the backbone parameters, denoted by $f_{n,\phi}(\mathcal{D}_n; \phi)$, and a separate objective on the head parameters, denoted by $f_{n,\psi}(\tilde{\mathcal{D}}_{n,\phi}; \psi)$, where,

$$\tilde{\mathcal{D}}_{n,\phi} = \{(z_i^n, y_i^n)\}_{i=1}^{|\mathcal{D}_n|} = \{(h_\phi(x_i^n), y_i^n)\}_{i=1}^{|\mathcal{D}_n|} \tag{4}$$

, i.e., the dataset $\mathcal{D}_n$ with each input sample $x_i^n$ being replaced by its low-dimensional representation $z_i^n = h_\phi(x_i^n)$. Then, the global backbone objective would be a variation of (2), where the end-to-end objectives are replaced by their backbone counterparts, i.e.,

$$\phi^* = \arg \min_{\phi \in \Phi} \frac{1}{N} \sum_{n=1}^N f_{n,\phi}(\mathcal{D}_n; \phi). \tag{5}$$

Similarly to (3), in order to derive the optimal backbone parameters $\phi^*$ using SGD, the backbone parameters at each client $n \in [N]$ can be sequentially updated as

$$\phi_{t+1}^n = \begin{cases} \phi_t^n - \eta \hat{\nabla}_\phi f_{n,\phi}(\mathcal{D}_n; \phi_t^n) & \text{if } t \bmod \tau \neq 0 \\ \frac{1}{N} \sum_{n=1}^N \phi_t^n & \text{o.w.} \end{cases} \tag{6}$$

Once the optimal backbone parameters $\phi^*$ are derived, each client $n \in [N]$ can freeze its backbone and train its personalized head parameters $\psi_n$ based on its local dataset $\tilde{\mathcal{D}}_{n,\phi^*}$, i.e.,

$$\psi_n^* = \arg \min_{\psi \in \Psi} f_{n,\psi}(\tilde{\mathcal{D}}_{n,\phi^*}; \psi). \tag{7}$$

### 2.2 RATE-DISTORTION THEORY AND MAXIMAL CODING RATE REDUCTION

Among the many ways to define the backbone objective $f_\phi(\mathcal{D}; \phi)$ to learn low-dimensional representations for a given dataset $\mathcal{D}$ (see, e.g., (Chen et al., 2020; Grill et al., 2020; Wang & Isola, 2020; Zbontar et al., 2021; Bardes et al., 2021), Lezama et al. (2018)), the *maximal coding rate reduction* (or, MCR$^2$, in short) has been recently proposed by Yu et al. (2020) as a theoretically-grounded way

of training low-dimensional representations based on the rate-distortion theory (Cover & Thomas, 2006).

Consider an i.i.d. sequence $\{z_i\}_{i \in [M]}$ of $M$ random variables following a distribution $p(z), z \in \mathcal{Z}$ and a distortion function $\omega : \mathcal{Z} \times \mathcal{Z} \to \mathbb{R}_+$. For a given $\Omega \geq 0$, the rate-distortion function is defined as the infimum $r$ for which there exist an encoding function $g_{\text{enc}} : \mathcal{Z}^M \to [2^{Mr}]$ and a decoding function $g_{\text{dec}} : [2^{Mr}] \to \mathcal{Z}^M$, such that

$$\lim_{M \to \infty} \frac{1}{M} \sum_{i=1}^{M} \mathbb{E}\left[\omega(z_i, \hat{z}_i)\right] \leq \Omega, \tag{8}$$

where the sequence $\{\hat{z}_i\}_{i \in [M]}$ denotes the reconstruction of the original sequence $\{z_i\}_{i \in [M]}$ at the decoder output, i.e.,

$$\{\hat{z}_i\}_{i \in [M]} = g_{\text{dec}} \circ g_{\text{enc}}\left(\{z_i\}_{i \in [M]}\right). \tag{9}$$

Intuitively, the rate-distortion function represents the minimum number of bits required to compress a given random variable, such that the decompressing error is upper-bounded by a constant $\Omega$.

In general, deriving the rate-distortion function is challenging, as it entails computing mutual information terms between the input sequence and the reconstructed sequence. However, for the case of finite-sample zero-mean multivariate Gaussian distribution with a squared-error distortion measure, the rate-distortion function has a closed-form solution. In particular, letting $Z = \begin{bmatrix} z_1 & \cdots & z_M \end{bmatrix} \in \mathbb{R}^{d \times M}$ denote the matrix containing a set of $M$ $d$-dimensional samples, for a squared-error distortion of $\epsilon^2$, the rate-distortion function is given by $\left(\frac{M+d}{2}\right) \log \det \left(I + \frac{d}{M\epsilon^2} ZZ^T\right)$, where $I$ denotes the $d \times d$ identity matrix (Ma et al., 2007). Quite interestingly, the rate-distortion function, when normalized by the number of samples, can be viewed as a measure of *compactness* of the given samples in $\mathbb{R}^d$. Assuming $M \gg d$, this leads to the *coding rate* $R(Z, \epsilon)$, defined as

$$R(Z, \epsilon) := \frac{1}{2} \log \det \left(I + \frac{d}{M\epsilon^2} ZZ^T\right). \tag{10}$$

The coding rate in (10) can be leveraged in a representation learning setup, where $z_i$'s are the representations produced by the backbone $h_\phi$. For representations to be useful, the representations within one class should be as compact as possible, whereas the entire set of representations should be as diverse as possible. For a given class $k \in [K]$, let $\Pi_k \in \mathbb{R}^{M \times M}$ be a diagonal binary matrix, whose $i^{\text{th}}$ diagonal element is 1 if and only if the $i^{\text{th}}$ samples belongs to class $k$. Then, the average per-class coding rate given the partitioning $\mathbf{\Pi} = \{\Pi_k\}_{k \in [K]}$ can be written as

$$R^c(Z, \epsilon | \mathbf{\Pi}) := \frac{1}{2M} \sum_{k \in [K]} \text{tr}(\Pi_k) \log \det \left(I + \frac{d}{\text{tr}(\Pi_k)\epsilon^2} Z \Pi_k Z^T\right), \tag{11}$$

where $\text{tr}(\cdot)$ represents the trace operation.

The principle of maximal coding rate reduction (MCR$^2$) proposed by Yu et al. (2020) defines the backbone objective $f_\phi(\mathcal{D}; \phi)$ as the difference between the average per-class coding rate $R^c(Z, \epsilon | \mathbf{\Pi})$ in (11) and the average coding rate over the entire dataset, $R(Z, \epsilon)$ in (10). More precisely,

$$f_\phi(\mathcal{D}; \phi) = -\Delta R(Z(\mathcal{D}; \phi)) = R^c(Z(\mathcal{D}; \phi), \epsilon | \mathbf{\Pi}) - R(Z(\mathcal{D}; \phi), \epsilon), \tag{12}$$

where the dependence of the representations $Z$ on the dataset $\mathcal{D}$ and the set of backbone parameters $\phi$ is explicitly shown. [1]

## 3 PROPOSED METHOD

Learning a low-dimensional representation can be posed as a collaborative objective, where each client in the network benefits from the collaboration. In federated learning, the dataset $\mathcal{D}$ is distributed among a set of clients, i.e., $\mathcal{D} = \cup_{n \in [N]} \mathcal{D}_n$, where $\mathcal{D}_n$ is the dataset located at the $n^{\text{th}}$ client.

---

[1] Since the MCR$^2$ backbone objective in (12) is monotonically decreasing with scaling the representations $Z$, in practice, the representations need to be constrained, e.g., to the unit hypersphere $\mathbb{S}^{d-1}$, or the Frobenius norm of per-class representations should be bounded by the number of per-class samples.

We leverage the MCR$^2$ principle to introduce the global objective of our proposed FL method, which we refer to as Federated Low-Dimensional Representation Learning, or FLOW, as follows,

$$\min_\phi f_\phi(\mathcal{D};\phi) := \frac{1}{2M} \sum_{k\in[K]} \log\det \left( I + \frac{d}{|\mathcal{M}_k|\epsilon^2} \sum_{n\in[N]} \sum_{m\in\mathcal{D}_n\cap\mathcal{M}_k} h_\phi(x_m)h_\phi(x_m)^T \right)$$
$$- \frac{1}{2}\log\det \left( I + \frac{d}{M\epsilon^2} \sum_{n\in[N]} \sum_{m\in\mathcal{D}_n} h_\phi(x_m)h_\phi(x_m)^T \right), \quad (13)$$

where for a given class $k \in [K]$, $\mathcal{M}_k$ denotes the set of samples that belong to the $k^{\text{th}}$ class. Note that in (13), we have made the dependency of the objective function on $\phi$ explicit, that is $z_m = h_\phi(x_m)$. It is worth noting that the objectives $f_\phi(\mathcal{D};\phi)$ in (12) and (13) are equivalent, as

$$Z = [z_1 \quad \dots \quad z_M] = [h_\phi(x_1) \quad \dots \quad h_\phi(x_M)], \text{ and } ZZ^T = \sum_{m\in[M]} z_m z_m^T, \quad (14)$$

and the partition matrix $\Pi_k$ has its $m^{\text{th}}$ diagonal element equal to one if and only if the $m^{\text{th}}$ belongs to $\mathcal{M}_k$. Therefore, learning low-dimensional representations in a distributed manner is equivalent to solving (13).

Note that as opposed to common FL implementations, our approach optimizes a common objective, as opposed to a summation over different objectives. However, this comes at a cost; the objective in (13) is not separable, i.e., it does not immediately follow that each client can take local gradient descent steps. In what follows, we will demonstrate interesting properties of problem (13), namely (i) that it is in each client's self interest to obtain a collaborative solution, and (ii) that a solution to problem (13) can be found in a distributed manner without clients needing to share their local datasets with each other.

### 3.1 MOTIVATION

Learning low-dimensional representations is a collaborative objective, and it is in each client's self interest to obtain a better representation. The choice of maximizing the coding rate reduction is well motivated by properties of the solution of problem (13), as can be shown in the following theorem.

**Theorem 1.** *Consider a set of dimensions $\{d_k\}_{k=1}^K$ such that with $rank(\mathbf{Z}_k^*) \leq d_k$. If the embedding space is large enough, i.e., $d \geq \sum_{k=1}^K d_k$ , and the coding precision is high enough, i.e. $\epsilon^4 < \min_{k\in[K]} \frac{|\mathcal{M}_k|d^2}{Md_j^2}$ then:*

- *The optimal subspaces associated with each class are orthogonal even from data across clients, i.e., $h_{\phi^*}(x_m)^T h_{\phi^*}(x_{\tilde{m}}) = 0$ for any $m \in \mathcal{M}_k, \tilde{m} \in \mathcal{M}_{\tilde{k}}$ with $k \neq \tilde{k}$; and,*

- *Each class subspace $Z_k^* = \sum_{m\in\mathcal{M}_k} h_{\phi^*}(x_m)h_{\phi^*}(x_m)^T$ achieves its maximal dimension $rank(Z_k^*) = |\mathcal{M}_k|$, and the largest $|\mathcal{M}_k| - 1$ singular values of $Z_k^*$ are equal.*

*Proof.* The proof follows from (Yu et al., 2020, Theorem 2.1) noting that problem (13) is equivalent to optimizing the centralized objective (12). A similar proof can also be found in (Chan et al., 2022, Theorem 1). □

Theorem 1 is important because it shows that the benefits of our method are two-fold: (i) the solution of the problem is orthogonal between classes, even from data coming from different clients, and (ii) the obtained representations for each class are maximally diverse. Theorem 1 is notable given that we are not sharing data between clients, and we are still able to learn representations that are orthogonal between classes. That is to say, if two samples $x \in \mathbb{R}^D$ and $x' \in \mathbb{R}^D$ belong to different classes, their corresponding low-dimensional representations $z$ and $z'$ will be orthogonal *regardless* of which client owns the datum. What is more, the subspace associated with class $j$ is maximal across clients, which translates into having a rich and diverse representation, even in low dimensions.

Note that if clients were to solve the problem individually, there would be two undesirable properties. First, even if the representations of samples of different classes for a given client are orthogonal,

---

**Algorithm 1** `FLOW`: Federated **LOW** Dimensional Representation Learning

---

1: Set coding precision $\epsilon$, step size $\eta$, embedding space dimensionality $d$, aggregation period $\tau$.
2: Initialize backbone parameters $\phi_0$.
3: **for** round $t = 1$ to $T$ **do**
4:     **if** $t \mod \tau \neq 0$ **then**
5:         **Client $n$ does:** Update model locally,

$$\phi_t^n = \phi_t - \eta \nabla_\phi f_\phi(\mathcal{D}_n; \phi_t^n),$$

        with $f_\phi$ given in (12).
6:     **else**
7:         **Server does:** Average models: $\phi_{t+1} = \frac{1}{N} \sum_{n=1}^{N} \phi_t^n$.
8:     **end if**
9: **end for**

---

that orthogonality might be violated when we move across clients, since there is no guarantee that per-class subspaces are aligned across clients. Therefore, having a common representation is a desirable property as it will enforce orthogonality between samples that do not co-exist at the same client. Second, the fact that the class subspace achieves its maximal dimension makes the representations more diverse, grouping similar samples together. Again, this property is desirable, and collaborating between clients is in each client's best interest. Note that these properties are properties of a centralized approach Yu et al. (2020), which our proposed method inherits and maintains in the distributed setting.

## 3.2 ALGORITHM CONSTRUCTION

The optimization problem in (13) is non-separable between clients, that is to say, the global objective is not equal to a summation, or an average, of individual objectives. Given that obtaining a closed-form solution of $\phi$ cannot be done in practice, we turn into an iterative SGD-based procedure. In short, at each round $t$, each client receives the current state of the model $\phi^t$, and utilizes its own data to maximize its own MCR$^2$ loss, as follows,

$$\phi_{t+1}^n = \phi_t - \eta \nabla_\phi f_\phi(\mathcal{D}_n; \phi_t), \tag{15}$$

with $\eta$ being a non-negative step size. Every $\tau$ rounds, the clients communicates their backbone parameters back to the central server. The central server's job is to average the received backbone parameters. Notice that these framework has two advantages: (i) clients do not need to share any of their private data, (ii) the computing is done at the edge, on the clients. Moreover, averaging the models between the clients can be done utilizing Homomorphic Encryption (HE), preventing the central client from revealing clients' gradient information. An overview of our proposed method can be found in Algorithm 1.

## 3.3 CONVERGENCE OF FLOW

In this section we analyze the convergence of `FLOW` (cf. Algorithm 1). To do so, we require the following assumptions,

**Assumption 1.** *The MCR$^2$ loss is G-smooth with respect to the parameters $\phi$, i.e.,*

$$\|\nabla_\phi f_\phi(\mathcal{D}_n; \phi_1) - \nabla_\phi f_\phi(\mathcal{D}_n; \phi_2)\| \leq G\|\phi_1 - \phi_2\|. \tag{16}$$

Assumption 1 is a standard assumption for learning problems. What this assumption implies is smoothness on the gradient of the function with respect to the parameters $\phi$. In the case of neural networks as the parameterization, this is a mild assumption, given the continuity of the non-linearity and its linear filters.

**Theorem 2.** *Consider the iterates generated by Algorithm 1. Under Assumption 1, if the client gradients are homogeneous unbiased estimates of $\nabla_\phi f_\phi(\mathcal{D}; \phi)$, i.e. $\mathbb{E}_{\mathcal{D}_n}[\nabla_\phi f_\phi(\mathcal{D}_n; \phi)] = \nabla_\phi f_\phi(\mathcal{D}; \phi)$, and the variance of the estimates of the gradients is bounded, i.e. $\mathbb{E}[\|\nabla_\phi f_\phi(\mathcal{D}_n; \phi) -$*

$\nabla_\phi f_\phi(\mathcal{D}; \phi)\|^2] \leq \sigma^2$, *then*

$$\frac{1}{T} \sum_{t=1}^{T} \|\nabla_\phi f_\phi(\mathcal{D}; \phi)\|^2 \leq \frac{G}{T}\left(f_\phi(\mathcal{D}_n; \phi_0) - f_\phi(\mathcal{D}_n; \phi_T)\right) + \frac{\sigma^2}{2N}, \tag{17}$$

*with* $\eta \leq 1/L$.

*Proof.* See Appendix A. ☐

If datasets $\mathcal{D}_n$ are composed of samples that are sufficiently similar, individual gradients taken at each client can be modeled as unbiased gradients of the gradients taken over the whole dataset, i.e., $\mathbb{E}_{\mathcal{D}_n}[\nabla_\phi f_\phi(\mathcal{D}_n; \phi)] = \nabla_\phi f_\phi(\mathcal{D}; \phi)$. Theorem 2 provides a standard convergence result for the case of a non-convex loss, which indicates that the summation of the norm of the gradient square does not diverge. The convergence of the summation implies that the norm of the gradient is in fact decreasing, which means that the iterates of the algorithm are approaching a first order stationary point.

We can also provide a proof of convergence of our algorithm in the case in which the distributions are not uniform in the clients.

**Theorem 3.** *Consider the iterates generated by Algorithm 1. Under Assumption 1, if the client gradients are a biased estimate of* $\nabla_\phi f_\phi(\mathcal{D}; \phi)$, *i.e.* $\mathbb{E}[\nabla_\phi f_\phi(\mathcal{D}_n; \phi)] = \nabla_\phi f_\phi(\mathcal{D}; \phi) + \mu_n$, *with* $\|\mu_n^T \nabla_\phi f_\phi(\mathcal{D}; \phi)\| \leq \delta$, *and* $\mathbb{E}[\|\nabla_\phi f_\phi(\mathcal{D}; \phi) - \nabla_\phi f_\phi(\mathcal{D}_n; \phi)\|^2] \leq \delta^2 + \sigma^2$, *then*

$$\frac{1}{T} \sum_{t=1}^{T} \|\nabla_\phi f_\phi(\mathcal{D}; \phi)\|^2 \leq \frac{G}{T}\left(f_\phi(\mathcal{D}; \phi_0) - f_\phi(\mathcal{D}; \phi_T)\right) + \frac{\sigma^2}{2N} + \delta, \tag{18}$$

*with* $\eta \leq 1/L$.

*Proof.* See Appendix B. ☐

Theorem 3 provides a convergence result of Algorithm 1 in the case of non-uniform clients. We model the non-uniformity of the client distributions by introducing a $\mu_n$ discrepancy vector for each client $n$. Notice that the key difference between Theorems 2 and 3 is the presence of $\delta$, which is a bound on the maximum norm of the discrepancy between the gradients. The consequence of such a dissimilarity is mild, as we can still obtain a convergent sequence.

## 4 EXPERIMENTS

We run our Algorithm 1 in two federated learning settings, with $N = 50$, and with $N = 100$ agents, in both cases, we run full participation, i.e. all agents were part of the communication rounds. For the dataset, we utilized CIFAR 10, and for the parameterization, ResNet18. The low dimensional representation has dimension $d = 128$. To model the agent mismatch, we distributed the samples per class according to a Dirichlet distribution prior with $\alpha = 5$, this distribution is widely used in the literature Shen et al. (2021); Hsu et al. (2019); Acar et al. (2021). In all cases we run for 500 epochs, with a learning rate of 0.3, we utilized a batch size of 500 samples, and we run 5 local epochs per agent.

### 4.1 LEARNING CURVES

In figure 1 we plot the learning curves for the MCR$^2$, as well as the $R$ loss, and the $R^C$ loss. It can be seen that in all cases, the centralized MCR$^2$ parameterization outperforms the Federated learning case. This is expected, as distributing the datasets tends to have a negative effect on performance. The number of agents also affects the loss, as the parameterization is able to get a better performance on $N = 50$ than on $N = 100$. This has to do with the unbiasness of the local gradients, that as the number of clients increases, so does the bias term. In all, figure 1 shows that the MCR$^2$ loss can be learned in a distributed manner.

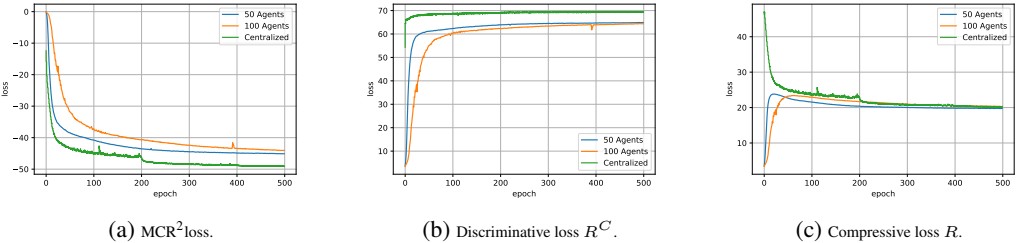

(a) MCR$^2$ loss.   (b) Discriminative loss $R^C$.   (c) Compressive loss $R$.

Figure 1: Learning curves for MCR$^2$ in Federated and Centralized settings for CIFAR-10.

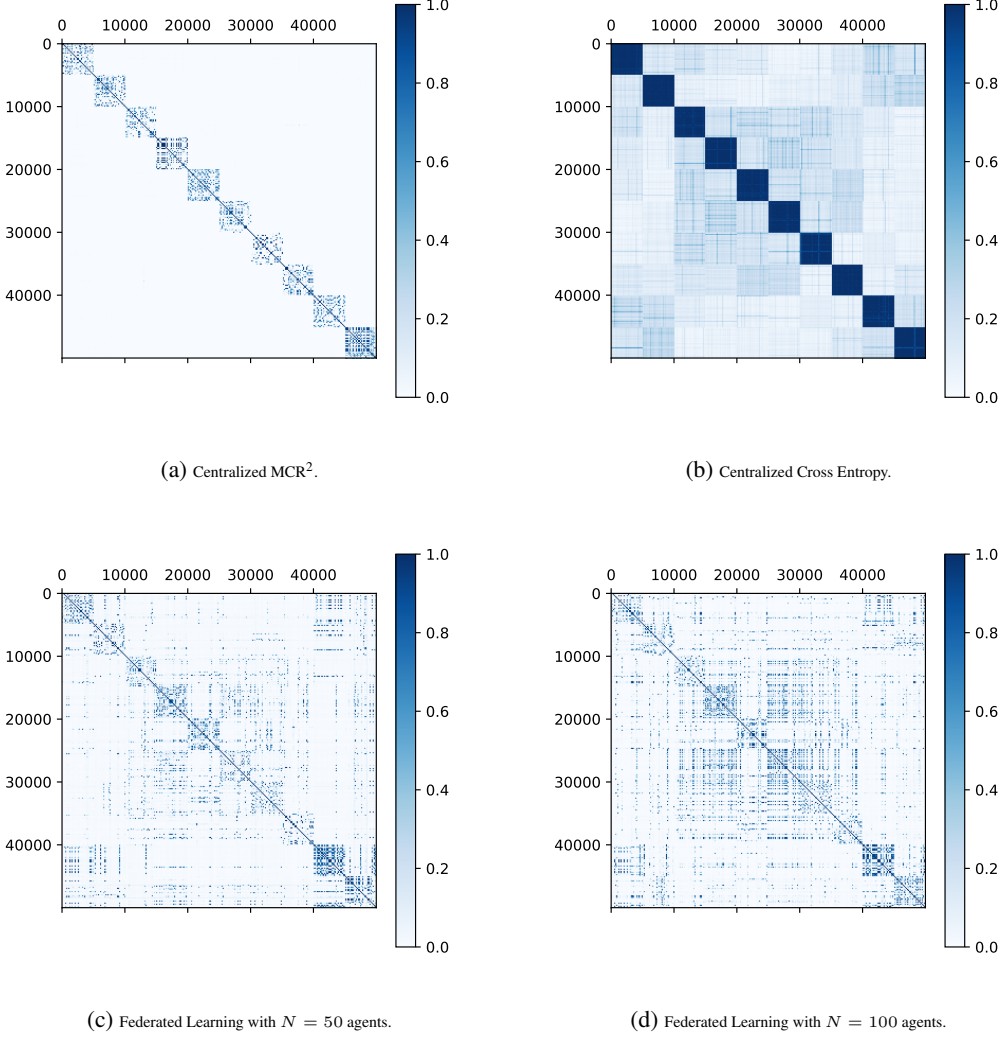

(a) Centralized MCR$^2$.   (b) Centralized Cross Entropy.

(c) Federated Learning with $N = 50$ agents.   (d) Federated Learning with $N = 100$ agents.

Figure 2: Orthogonality of the low dimensional representation.

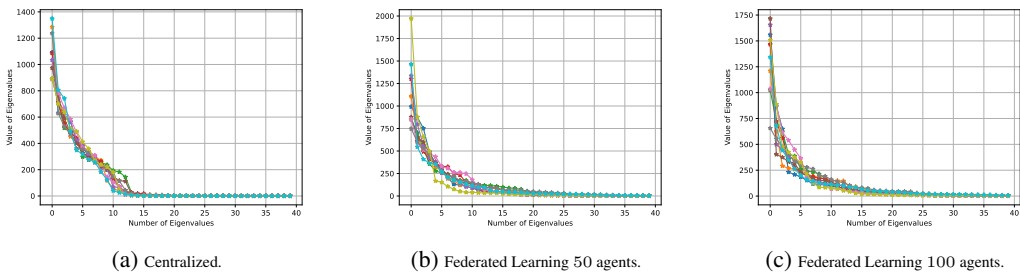

(a) Centralized.   (b) Federated Learning 50 agents.   (c) Federated Learning 100 agents.

Figure 3: Decreasing order of magnitude of singular values of the subspaces associated with each class.

## 4.2 ORTHOGONALITY OF REPRESENTATIONS

Figure 2 shows the cosine similarities between all the elements of the dataset. Upon training, we obtained the low dimensional representation of each sample, and computed the pairwise cosine correlation between them. In order to plot the samples, we ordered so that the first $10000$ samples belong to the first class and so on so forth. As expected by Theorem 1, samples of different classes tend to be orthogonal between themselves, and samples of the same class are maximally diverse. Consistently with the worse value of the loss observed in Figure 1, we can visually verify that the orthogonality between samples is worse as the number of clients increases. Nevertheless, for the most part, we are able to obtain an orthogonal representation for the samples. This, is as expected by Theorem 1, 2, 3. As opposed to the centralized case, in our federated learning procedure, samples of different agents are never shared, which adds merit to Figure 2. The value of using the MCR$^2$ as a loss is seen when compared to the representations learned with the cross entropy loss. To obtain this representation, we train a centralized architecture (i.e. ResNet 18) with $128$ features before the fully connected layer. Figure 2 shows that learning orthogonal representations is not obtained unless enforced. Moreover, the block diagonal elements of the cross entropy matrix are darker, which means that the numbers are closer to $1$. This comes to no surprise, as the sole objective of the cross-entropy loss is to separate samples of different classes. However, the MCR$^2$ loss also seeks for diverse representations, allowing samples of the same class to have different alignments.

Finally, Figure 3 shows the distribution of the eigenvalues of the per-class matrices $Z_k Z_k^T$ or the singular values of $Z_k$ for different classes in centralized and federated cases. Again, we see that our proposed approach can lead to similar distributions of the principal components of the learned representation subspaces, where each class ends up occupying a low-dimensional subspace, even though each client does not have direct access to the data samples hosted by other clients.

## 5 CONCLUSION

In this paper we introduced a principled procedure to learn low-dimensional representations in a distributed manner. In the context of Federated Learning, we introduce a collaborative loss based on the maximal coding rate reduction (MCR$^2$), which individually benefits all the agents in a self interested way. We refer to our federated low-dimensional representation learning algorithm by FLOW. Theoretically, we show that (i) the solution of FLOW generated orthogonal representations for samples of different classes, and maximizes the dimension of each class subspace, and (ii) that under mild conditions, FLOW converges to first order stationary point. Empirically, we compare our method to the centralized procedure, validating all the claims that we put forward.

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
