# OpenReview forum: "Federated Representation Learning via Maximal Coding Rate Reduction"
_ICLR.cc/2023/Conference — Submitted to ICLR 2023_

### Official Review · Reviewer_fuw1 · 2022-10-12

**Confidence:** 5
**Correctness:** 1
**Technical Novelty And Significance:** 1
**Empirical Novelty And Significance:** 1
**Recommendation:** 1

**Clarity, Quality, Novelty And Reproducibility:**

The novelty of the work is extremely limited. The writing becomes unclear when it comes to whether the normalization constraints are imposed on the MCR2 objective.

**Strength And Weaknesses:**

# Weakness
1. The work is not solid. My
   judgment is based on the following:
   - The background section is unnecessarily long. For example, the
     review of Section 2.2 is redundant, and the paper can simply say
     that, as motivated by some rate-distortion theory, Yu et al.
     (2020) proposed this MCR2 objective.
   - Theorem 1 is basically a copy-and-paste from a follow-up work
     (Ryan et al., 2022) of Yu et al. (2020), called ReduNet.
     Moreover, the copy is wrong: It misses the key assumption that the
     objective admits a low-rank solution, see Theorem 12 of Ryan et al.
     (2022). Finally, the paper did not mention that this theorem is
     due to Ryan et al. (2022).

   - Theorem 2 is a very
     standard result for smooth optimization. Moreover,
     footnote 1 of the paper has noted that the MCR2 objective
     requires some non-convex constraints for itself to be well-posed,
     but Theorem 2 (and perhaps also the implementation) simply
     ignored these constraints, which are required for Theorem 1. This
     creates a great mess.
   - Figures similar to Figure 2 have already been reported by Ryan et
     al. (2022). The purpose of such a repeat is not very clear.

2. It is quite confusing why the paper keeps saying that MCR2 is a
   principle (Similar objectives to MCR2 already exist, cf. Lezama et
   al., (CVPR 2018), why their objectives are not principles?). Also,
   the coding rate reduction explanation is only applicable to Gaussian data.
   And one gets no interpretation when dealing with real data. It is
   also not very clear to me why diverse representations are
   practically important, and the paper did not show the performance
   benefits compared to the cross-entropy loss.

3. Here are the references that I mentioned above:
   - https://arxiv.org/pdf/2105.10446.pdf (Ryan et al., 2022)
   - https://arxiv.org/abs/1712.01727 (Lezama et al., CVPR 2018)

**Summary Of The Paper:**

The paper considers the federated learning problem and proposes to
minimize the difference between two log determinant functions instead of
the cross-entropic loss.



**Summary Of The Review:**

As detailed above, the paper is clearly below the bar of ICLR in several
aspects. In its current form, the paper is better suited for some less
selective conferences.

---

> ### Author Response · Authors · 2022-11-14
> **Reply**
>
> Thank you for the insightful review, and the comments made, which we believe will help us improve the overall quality of our work. Below, we will address your comments in detail.
>
> **Length of Background Section:** We kindly appreciate your comment regarding the length of the Background Section, and we will revise this section to summarize the different points covered in it as much as possible. That being said, we believe that the length of the Background Section is standard for a machine learning conference. We believe that by simply referring to Yu et al. (2020), we will not be able to present the objective clearly and in a self-contained way. We would like to emphasize the fact that we want the paper to be self-inclusive and that the reader can follow the formulation without the need to resort to the rest of the literature on federated learning and information theory. We believe this section makes readers better appreciate the connections between these areas of research, which might seemingly be disconnected.
>
> **Comments on Theorem 1:** We respectfully disagree with suggestion that we are intentionally leaving out (Ryan et al., 2022) in our manuscript. We note that the Theorem was first presented in (Yu el al., 2020), and that is exactly how we prove it, as mentioned in Appendix A of the manuscript:
>
>
> >  The proof follows from (Yu et al., 2020, Theorem 2.1) noting that problem (13) is equivalent to optimizing the centralized objective (12).
>
> In our manuscript, we clearly state where the result and the proof originate from, and we cite it accordingly. We do not claim the result to be ours. As you also alluded to, (Ryan et al., 2022) is a follow-up work of Yu et al. (2020), with an essentially similar proof. That is why we chose to cite the former work. Having said that, following the reviewer's suggestion, we added the citations to (Ryan et al., 2022) and (Lezama et al., CVPR 2018). We have also moved the proof line to the main body of the paper. Moreover, we have modified the statement of the Theorem to mention the low rank of the subspaces $\{Z_k^*\}_{k=1}^K$.
>
> We would like to point out that the claim about this work being dishonest is far from what this work is doing. We never claim any result that is not ours to be ours. At all points throughput the paper, we refer the reader to prior work and we clearly establish on what existing work our manuscript is based on.
>
> **Comments on Theorem 2:** We thank the reviewer for this comment. Regarding the theoretical aspect of Theorem 2, note that the normalizing constraint is part of the function $f_\phi$ and, therefore, it is implicit in the parameterization. This is standard, as for example in the case of cross-entropy loss, the output of the function is a probability vector and, hence, it is also normalized.
>
> We respectfully believe that the comment "perhaps also the implementation" is not accurate. The implementation does include the projection. As can be seen in the supplementary material in `resnet_mcr.py`, in line 102, we normalize the output:
>
> >         return F.normalize(out)
>
>
>
> **Comments on Figure 2**: We thank the reviewer for this question. The purpose of Figure 2 is to show the quality of the representations learned distributively. We show that the MCR$^2$ as an objective renders solutions that satisfy what they are intended to. Note that in the case of Ryan et al., their experiments are in a centralized setting. In our case, we extend these results to a distributed setting.
>
>
> **Comments on MCR$^2$ being a principle**: We agree with the reviewer that MCR$^2$ is not the only principled way to learn high-quality representations, and we are not dismissing other existing objectives such as that of Lezama et al. (CVPR 2018). We have added this citation to the revised version of the manuscript.
>
>
> **MCR being applicable only to Gaussian data**: When optimizing the MCR$^2$ objective, we assume that the low-dimensional representation $z$ is Gaussian, not the input data $x$. The intuition for this assumption is that the distribution of real-world data of a class follows a Gaussian distribution in the low-dimensional embedding space. That is to say, optimizing this objective actually encourages the distribution of the embeddings to converge to a multi-variate Gaussian distribution.

---

> > ### Author Response · Authors · 2022-11-14
> > **Reply**
> >
> > **Benefit of diverse representations**: Diverse presentations are important because we can further utilize them for other tasks. It is useful to note that in the case where accuracy is the only end goal, then the end-to-end procedure should be done using the cross-entropy (CE) loss.  However, we are motivated by the fact that learning compact representations of data is much more meaningful (and harder) than just learning a pattern matching model to do classification, and these representations can be used in a variety of other settings, such as transferring to new datasets and tasks, comparing similarities/distances between samples, etc. The performance against the CE loss is done in Figure 2. Note that given that we are interested in diverse representations, the CE loss is not useful. As shown in Figure 2, the CE loss collapses all the representations within a class into a single vector, and is therefore unable to draw any distinctions between different samples belonging to the same class. That is to say, CE loss obtains representations that are useful for classifying samples into classes, but within a class, no other detailed information about the samples is obtained.

---

> > ### Comment · Reviewer_fuw1 · 2022-12-02
> > **Reply to Authors' rebuttal**
> >
> > Dear authors,
> >
> > thanks for the rebuttal. I appreciate it.
> >
> > **On Theorem 1**: I am retracting my statement on the dishonesty of Theorem 1. However, I believe that it is a simple standard practice to write directly in the main paper that "this is a direct consequence of Theorem X of Paper Y", and, in my opinion, Theorem 1 should be called a corollary.
> >
> > Sure, the authors did not claim the results are theirs, but the paper is theirs.
> >
> > **On Theorem 2**: Similarly, Theorem 2 is a basic corollary of some well-known results (or a very basic and well-known analysis of SGD), which the authors did not cite, mention, or appreciate. I would expect a sentence such as "Theorem 2 follows from a standard analysis of SGD". I am not convinced by the rebuttal which claims that it is standard to define the normalization constraint as part of the function. If that were the case then why there is a distinction between **constrained optimization** and **unconstrained optimization**?
> >
> > Let me point out a paper for the authors to copy:  "Global rates of convergence for nonconvex optimization on manifolds" (Boumal et al. 2018)
> >
> > The objective function under consideration is smooth, and the norm constraints define a smooth manifold. So I think it would be better to study and copy the theorems of (Boumal et al. 2018). Actually, I am sharing a novel idea here, because such a result is not in the papers of MCR2.
> >
> > Finally, I am not a politician and am not convinced by authors' argument on the benefit of diverse representations. When we do classification, we care about performance on classification.

---

### Official Review · Reviewer_LtxX · 2022-10-24

**Confidence:** 2
**Correctness:** 4
**Technical Novelty And Significance:** 1
**Empirical Novelty And Significance:** 1
**Recommendation:** 3

**Clarity, Quality, Novelty And Reproducibility:**

The paper is overall well-written and very readable. But it lacks novelty.


**Strength And Weaknesses:**

Strength:

The idea of separating the backdone network from the project head is an interesting idea.

Weakness:

The idea proposed seems rather trivial. And the empirical evaluation is not done in comparison with some natural baselines. It is difficult to see the significance of the proposed work.



**Summary Of The Paper:**

The authors proposed to replace the cross-entropy loss with the MCR2 loss in federated learning. They provided a theoretical analysis of learning convergence of the global model using the MCR2 loss. They also empirically show the convergence of the loss and the orthogonality of the class-wise subspaces learnt.

**Summary Of The Review:**

The paper makes a rather simple change to the standard federated learning setting by using a different loss function. The empirical evaluation does not show the advantage of performing that change.

---

> ### Author Response · Authors · 2022-11-14
> **Reply**
>
> Thank you for reviewing our paper in an insightful way. In what follows, we provide our responses to your comments.
>
> **Novelty of the proposed idea:** In your review, you argue that the idea of learning a low-dimensional representation in a distributed manner utilizing the MCR$^2$ principle is rather trivial. We would like to point out that most prior work in the federated learning literature focuses on learning classifiers in an end-to-end fashion. We leverage the importance of learning representations in FL and, therefore, propose to close this gap by proposing to learn a low-dimensional representation utilizing the MCR$^2$ principle.
>
> That is to say, our work observes that in the federated learning context, learning representation is transcendental. This is due to the fact that client heterogeneity is ubiquitous in federated learning, and, therefore, learning a common classifier renders sub-optimal solutions. To this end, we propose to jointly learn the representations, given that this is common to all agents. In this work, we propose the MCR$^2$ as the objective to learn low-dimensional representations, and we show that the solution to the MCR$^2$ loss attains representations that benefit *all* agents.
>
> While it is true that this work does not invent federated learning or MCR$^2$, we believe the combination of the two is a novel contribution to both the fields of representation learning and federated learning. We hope our answer helps to clarify the novelty of our work.
>
> **Baselines:** Thank you very much for pointing this out. We compare the performance with the centralized MCR$^2$, and show that the degradation caused by learning distributively is acceptable as it allows us to learn a representation that is orthogonal and maximally diverse. We believe that our simulations empirically demonstrate the theoretical claims that we put forward in our work. That being said, please let us know if you have specific suggestions on what experiments we could add to better showcase the benefits of our proposed method.

---

### Official Review · Reviewer_bnnY · 2022-10-25

**Confidence:** 3
**Correctness:** 3
**Technical Novelty And Significance:** 2
**Empirical Novelty And Significance:** 2
**Recommendation:** 5

**Clarity, Quality, Novelty And Reproducibility:**

The paper is well written and easy to understand and the code is released for the reproducibility. The novelty of the paper is quite limited.


**Strength And Weaknesses:**

Strength: The work is overall well-written and easy to read, all the details of the cocepts, setups are clearly stated. The theoretical results demonstrate that in the distributive setting,  the learning objective could still be successfully solved.

Weakness: The novelty of the work is quite limited, although the usage of the MCR loss in federated learning is new, both the loss and the algorithm design are not new. Besides, the experimental part is quite weak as it only demonstrates the results of the MCR loss under Federated Learning setting, no comparison has been done to show the supremacy of the proposed algorithm.


**Summary Of The Paper:**

The paper proposes an algorithm to learn low-dimensional representations in the Federated Learning setting. The paper borrows idea from a recent work which formulated a loss that could maximize the code rate difference between the entire dataset and summation of individual classes. The work takes a further step by identifying that learning a low-dimensional representation could be a collaborative objective and introduces an algorithm to solve the learning objective in a distributed manner.


**Summary Of The Review:**

This is an well-written paper proposing an interesting extension for the MCR loss in the federated learning setting. The experimental parts are a bit weak and the overall novelty is also limited.

---

> ### Author Response · Authors · 2022-11-14
> **Reply**
>
> Thank you very much for the detailed review of our paper, and the insightful comments on the work. Below, we address each of the raised points.
>
> **Novelty:** Our work observes that in the federated learning context, learning representation is transcendental. This is due to the fact that client heterogeneity is ubiquitous in federated learning, and, therefore, learning a common classifier renders sub-optimal solutions. To this end, we propose to jointly learn the representations, given that this is common to all agents. In this work, we propose the MCR$^2$ as the objective to learn low-dimensional representations, and we show that the solution to the MCR$^2$ loss attains representations that benefit *all* agents.
>
> While it is true that this work does not invent federated learning or MCR$^2$, we believe the combination of the two is a novel contribution to both the fields of representation learning and federated learning. We hope our answer helps to clarify the novelty of our work.
>
> **Numerical Experiments:** We kindly note that the experiments showcase that MCR$^2$ can be maximized in a distributed manner with heterogeneous datasets, and this empirically validates the claims that we put forward in the paper. That being said, please let us know if you have specific suggestions on what experiments we could add to better showcase the benefits of our proposed method.

---

### Official Review · Reviewer_L6k4 · 2022-10-26

**Confidence:** 5
**Correctness:** 4
**Technical Novelty And Significance:** 1
**Empirical Novelty And Significance:** 1
**Recommendation:** 3

**Clarity, Quality, Novelty And Reproducibility:**

Clarity is fine
Quality is low because of insufficient experiments.
Novelty is very limited.
Reproducibility is fine.

**Strength And Weaknesses:**

Strength:

S1). The paper is clearly presented. Basically, the paper is well-organized and easy to follow. The authors give sufficient background on both federated learning and MCR2.

S2). The references reviewed are somehow complete. The authors presented both traditional Federated learning and personalized FL.

Weakness:

W1). the work lacks novelty and the contribution is very limited. Basically, it is only a simple adoption of MCR2 to the federated setting. Since the optimization is also SGD and fedavg, there are no new challenges to transferring MCR2 objective to a federated setting including the proofs of convergence and others.

W2). Moreover, since the optimization needs to collectively sum over different labels (K), however, in the personalized FL setting, it is possible that the same sample in different clients is assigned with different labels, as mentioned in paper https://arxiv.org/pdf/1910.01991.pdf
In this case, how the proposed obj of MCR2 would work in FL setting?

W3). The representation learning framework MCR2 is still using class labels, why not use cross-entropy loss? what's the extra merits? The authors need to clarify the usage of such supervised embedding learning compared to popularly used cross-entropy loss with information bottleneck guidance.

W4). The experiments are far from sufficient. Basically, the authors only present the convergence and effectiveness of learning a compact representation learning. Basically, what are the usefulness and the quality of the learned representations? What's the superiority of the optimization framework fast converge at the FL setting compared to traditional FedAVG?



**Summary Of The Paper:**

The submission proposes a federated methodology to learn low-dimensional representations from a dataset distributed among several clients. Basically, the idea lacks novelty and is with limited contribution as it is a simple application of the MCR2 objective to a federated learning setting. The optimization is still using fedAVG and the proofs are also routine as FedAVG as the embedding parameters are optimized based upon SGD.

**Summary Of The Review:**

Since the work lacks novelty and the contribution is very limited, I thus do not champion acceptance. Basically, it is only a simple adoption MCR2 to the federated setting. Since the optimization is also SGD and fedavg, there are no new challenges to transferring MCR2 objective to a federated setting including the proofs of convergence and others.

---

> ### Author Response · Authors · 2022-11-14
> **Reply**
>
> Thank you very much for your insightful review of our paper. In what follows, we provide point-by-point responses to your comments.
>
> **Novelty of our work:** Our work observes that in the federated learning context, learning representation is transcendental. This is due to the fact that client heterogeneity is ubiquitous in federated learning, and, therefore, learning a common classifier renders sub-optimal solutions. To this end, we propose to jointly learn the representations, given that this is common to all agents. In this work, we propose the MCR$^2$ as the objective to learn low-dimensional representations, and we show that the solution to the MCR$^2$ loss attains representations that benefit *all* agents.
>
> While it is true that this work does not invent federated learning or MCR$^2$, we believe the combination of the two is a novel contribution to both the fields of representation learning and federated learning. We hope our answer helps to clarify the novelty of our work.
>
> **Samples shared between classes:** Theoretically, we will not be able to obtain orthogonal low-dimensional representations in the case where samples belong to more than one class. However, in practice, there might be some overlap between classes in the low-dimensional representation, and if the number of samples that have mismatch labels between agents is small, these samples will be assigned to a small number of overlapping sub-spaces.
>
> **Use of labels to find low-dimensional representations:** In the case of federated learning, it is a common assumption to know the labels of the data, and, therefore, we believe that this information should be used in our benefit. We believe that MCR$^2$ is a well-suited objective to learn low-dimensional representations for labeled data. That being said, we agree that extending MCR$^2$ to unsupervised/self-supervised settings is an important future direction, which we leave for future work.
>
> **Numerical experiments:** The numerical results showcase that MCR$^2$ can be maximized in a federated learning setting, and, therefore, representations can be learned as opposed to just learning the classifiers. The numerical examples that we put forward show that representations can be learned in a distributed way, and are successful in that regard. Please let us know if you have suggestions on what experiments we could add to better showcase the benefits of our proposed method.

---

### Decision · Program_Chairs · 2023-01-20

**Decision:**

Reject

**Justification For Why Not Higher Score:**

Insufficient evaluation to support all of the claims made, and limited novelty.

**Justification For Why Not Lower Score:**

N/A

**Metareview: Summary, Strengths And Weaknesses:**

This paper proposes an approach to federated representation learning based on the so-called principle of maximal coding rate reduction. The approach is connected to personalized federated learning, and the aim is to learn a shared backbone (feature extractor) across clients, while clients learn a personalized prediction head. The paper claims to contribute theory ensuring that optimal subspaces will be learned under certain assumptions, and that the proposed federated method FLOW converges. Experiments are reported for simulations of federated training of a ResNet18 on CIFAR10.

The proposed approach may be promising based on the CIFAR10 results.

Overall, there were several weaknesses noted about this work. The main weakness is the limited empirical evaluation; the paper could be substantially strengthened by exploring this method on several datasets in several settings, to both illustrate the advantages over other personalized FL methods, and also to identify potential limitations. Several reviewers questioned the novelty and expressed the opinion that just applying the idea of MCR2 loss in the context of federated learning brought limited novelty.

**Summary Of Ac-Reviewer Meeting:**

n/a